# Leg Dominance—Surface Stability Interaction: Effects on Postural Control Assessed by Smartphone-Based Accelerometry

**DOI:** 10.3390/sports11040075

**Published:** 2023-03-30

**Authors:** Arunee Promsri, Kotchakorn Bangkomdet, Issariya Jindatham, Thananya Jenchang

**Affiliations:** 1Department of Physical Therapy, School of Allied Health Sciences, University of Phayao, Phayao 56000, Thailand; 2Unit of Excellence in Neuromechanics, School of Allied Health Sciences, University of Phayao, Phayao 56000, Thailand

**Keywords:** neuromuscular control, injury prevention, laterality, unstable surface, sample entropy

## Abstract

The preferential use of one leg over another in performing lower-limb motor tasks (i.e., leg dominance) is considered to be one of the internal risk factors for sports-related lower-limb injuries. The current study aimed to investigate the effects of leg dominance on postural control during unipedal balancing on three different support surfaces with increasing levels of instability: a firm surface, a foam pad, and a multiaxial balance board. In addition, the interaction effect between leg dominance and surface stability was also tested. To this end, a tri-axial accelerometer-based smartphone sensor was placed over the lumbar spine (L5) of 22 young adults (21.5 ± 0.6 years) to record postural accelerations. Sample entropy (SampEn) was applied to acceleration data as a measure of postural sway regularity (i.e., postural control complexity). The results show that leg dominance (*p* < 0.001) and interaction (*p* < 0.001) effects emerge in all acceleration directions. Specifically, balancing on the dominant (kicking) leg shows more irregular postural acceleration fluctuations (high SampEn), reflecting a higher postural control efficiency or automaticity than balancing on the non-dominant leg. However, the interaction effects suggest that unipedal balancing training on unstable surfaces is recommended to reduce interlimb differences in neuromuscular control for injury prevention and rehabilitation.

## 1. Introduction

The preference for performing unimanual motor tasks with one leg (i.e., leg dominance) is a well-established trait of human motor control resulting from the inherent bilateral asymmetry between the motor control circuitry of the two hemispheres [1]. Furthermore, leg dominance has been reported to be clinically associated with a prevalence of sports-related lower-limb injuries (e.g., an anterior cruciate ligament (ACL) injury) [2,3,4]. Postural control, the ability to control the body’s position in space in order to maintain stability and orientation [5], is one of the physical performance abilities selected to determine the interlimb difference. However, as previously reported in systematic review studies, the effects of leg dominance, commonly measured by assessing posturography, e.g., center-of-pressure (COP) oscillations through several COP-based variables, are challenging to detect in postural control [6,7]. In this sense, rather than determining COP motions, other biological signals, e.g., postural acceleration, were investigated primarily among physically active young adults. For example, recent studies have investigated neuromuscular control in terms of the postural acceleration of individual movement components (i.e., movement strategies), decomposed from the whole-body postural movement during unipedal balance tasks [8,9,10]. Their results revealed that differences between the dominant and non-dominant legs, shown when performing unipedal balancing on stable surfaces [8], disappeared when balancing on unstable support surfaces [9,10]. Based on previous findings [8,9,10], exercising or training on an unstable surface may help to reduce interlimb differences in postural control.

Every human movement has been accepted to result from acceleration (i.e., muscle-driven acceleration) [11,12]. For instance, in the case of unperturbed postural control, it has been proposed that the postural accelerations are either (I) a direct result of multiple muscle actions [12], (II) a result of the neuromuscular system using gravity to produce desired acceleration by reducing muscle activation [13], or (III) an unintended result of gravity that the neuromuscular system is unable to prevent, e.g., losing stability due to insufficient muscle action [11,13]. However, in order to prevent segmental or intersegmental acceleration, the neuromuscular system may stimulate muscles by co-activating muscle groups to tighten joints [14]. In this regard, postural acceleration has been suggested as one of the most critical mechanical variables, reflecting the ability of the sensorimotor system to control the body’s motion and maintain its stability [11,12]. Hence, assessing laterality effects through acceleration-based variables is of interest, not only from a mechanical point of view, but particularly from the perspective of the inherent ability of neuromuscular control [11], which may provide relevant information for injury prevention and rehabilitation.

Postural control is believed to result from nonlinear interactions among multiple neuromuscular elements as well as internal and external factors [15,16]. Using nonlinear analyses, e.g., sample entropy (SampEn) [15,17,18,19,20,21], that consider the temporal evolution of postural adjustments effectively provides information about changes in postural control caused by various factors. For example, SampEn is a variable which is widely used to estimate the inherent complexity of the postural control system by measuring the regularity (predictability) of postural stability, such as that it is detected by trunk acceleration [22], with low entropy for predictable (e.g., periodic) signals and high entropy for unpredictable signals [18,21]. High SampEn values (high irregularity) of postural sway indicate that the postural control system controls the movements in a more variable or adaptable manner [18,21]. In this sense, a more irregular postural sway fluctuation (high SampEn) emerged while balancing tasks, reflecting an improvement in postural control efficiency or automaticity [18,21,23]. On the other hand, a more regular postural sway pattern (low SampEn) has been interpreted as an ineffective postural control or more rigid postural behavior [18,21,23]. Moreover, it may be concluded that if SampEn is too high, the movements will become too irregular and random. Ideally, SampEn should be somewhere in the middle of its theoretical range, with deviations from this in either direction reflective of poorer postural control.

Although many gold-standard instruments have been proposed for taking quantitative balance measurements, the cost and complexity of the devices for assessing quantitative data may make it difficult to evaluate and interpret results and limit academic research or clinical settings [24]. As a result, less expensive sensors have been considered alternatives for balancing evaluations. For instance, smartphones and other gadgets with incorporated triaxial accelerometers and gyroscopes are innovative technologies that can evaluate balance based on acceleration-measured data, which turns these devices into wireless inertial measurement units (IMU) [24,25,26,27]. The most common way to describe balance and sway displacement is to use the represented center-of-mass (COM) sway, estimated as a single point around the base of the lumbar spine, which is often selected as a way to attach the smartphone and other gadget-based accelerometry [25,28]. As previously reported, smartphone-based accelerometry is a valid and reliable measure of postural stability, one which may be variously applied to detecting fall risk in older adults [26] and measuring postural control in patients with Parkinson’s disease [29] and multiple sclerosis [30]. 

In summary, the current study aimed to investigate the effect of leg dominance on postural control in young adults performing non-exercise physical activity when unipedally balancing on three different support surfaces, each with increasing instability, by assessing postural acceleration using a smartphone sensor. In addition, the effects of surface stability and the interaction between leg dominance and surface stability were also tested. Based on the previous findings that leg dominance effects disappeared when performing unipedal balancing on an unstable surface [9], it was hypothesized that the interaction between leg dominance and surface stability would be observed, especially in measuring SampEn, due to the nature of postural control as a result of nonlinear interactions between internal (e.g., neuromuscular control) and external (e.g., environment) factors. 

## 2. Materials and Methods

### 2.1. Participants

Twenty-two young adults (13 females and 9 males), who generally had daily physical activity but no specific sports or exercise participation, participated in the study (Table 1). This group of participants with non-exercise physical activity was selected due to several previous research studies in physically active young adults showing the influence of leg dominance on postural control when studying postural acceleration [8,9,10]. Therefore, focusing on non-exercise physical activity may provide more information regarding laterality effects on postural control. The sample size of the current study was calculated using a priori power analysis through the G*Power software (Heinrich-Heine-Universität Düsseldorf, Düsseldorf, Germany) [31] based on two previous studies which assessed the interlimb differences in unipedal postural control by measuring postural acceleration [9,10], yielding an average effect size (Partial Eta Square; η_p_^2^) for comparisons between the dominant and non-dominant legs of 0.37. Based on this computation, with a significant level of a = 0.05 and a desired power = 0.95, the suggested sample size was N = 18. However, twenty-two young adults volunteered to participate in the current study. All participants self-reported no neurological and musculoskeletal problems and had received no balance-specific training within the last six months. The right leg of all participants was the dominant leg, as determined by the preferred leg for kicking a ball [8], which coincided with their dominant hand, as determined by the writing hand. All experimental procedures were performed with the approval of the Institutional Review Board of the University of Phayao, Thailand (Approval Code No.: UP-HEC 1.2/047/65) and in accordance with the Declaration of Helsinki. All volunteers provided written, informed consent before their participation.

### 2.2. Equipment

Tri-axial accelerometry was measured using a smartphone (OPPO A92, Guangdong Oppo Mobile Telecommunications Corp., Ltd., Dongguan, China). The Physics Toolbox Sensor Suite application (version 2022.09.11), found on the Google Android platform, was used to collect and export the acceleration data at a sampling rate of 200 Hz generated by the smartphone accelerometer [32]. The center of the smartphone was horizontally placed on the lumbar region (L5) using a waist-mounted pouch close to the body’s COM [25,28]. 

Three different support surfaces were used in the current study (Figure 1): a firm surface, a foam pad (Airex Balance Pad Elite, Alcan Airex AG, Sins, Switzerland), and an MFT Challenge Disc (Trend Sport Trading GmbH, Großhöflein, Austria). A foam pad is a rectangular piece of blue foam measuring 50 cm in width, 41 cm in length, and 6 cm in height, while an MFT Challenge Disc is a multiaxial balance board consisting of an upper 44 cm diameter round plate connected to a base circle plate by a group of four rubber cylinders with an 8 cm height at the middle of the plate. 

### 2.3. Experimental Procedure

All participants initially started with a 15 s familiarization trial on the foam pad and the balance board without instruction or feedback. Then, on each foot, volunteers performed three 60 s balancing trials barefooted on three different support surfaces: a firm surface, a foam pad, and an MFT Challenge Disc. The order of the tests (right foot–left foot/left foot–right foot) and the order of the surface conditions were randomized. Three trials of balancing were conducted for each foot and each surface. 

In order to standardize the starting position, each participant was asked to place their hands on the iliac crests, to unipedally barefoot stand by keeping the base of the second metatarsal bone over a reticle crossline marked on each surface (Figure 1), and to flex the hip and knee of the lifted leg at 20 and 45 degrees, respectively [8]. During testing, all volunteers were asked to focus their gaze on a 10 cm diameter black circle placed on the wall approximately two meters away at individual eye level, to stand still for the firm and foam surface conditions, to try keeping the balance board horizontal for the MFT conditions, not to touch the stance leg with the lifted leg, and to avoid any movements (e.g., scratching) not required for balancing [8,10]. If the participant lost their balance by touching the floor with the lifted leg, a new test was required. After each test, participants could rest for one to three minutes at their discretion but were not allowed to stand on the unstable platform during the rest time.

### 2.4. Data Analysis

All data processing was performed using MATLAB^TM^ (MathWorks Inc., Natick, MA, USA). A Fourier analysis was performed on the raw acceleration signals, revealing that the highest power resided at frequencies around 5–10 Hz, but that visible power was still observed in the frequency range between 15 and 20 Hz. Therefore, these signals were filtered with a 4th-order zero-phase 20 Hz low-pass Butterworth filter, as similarly reported in a previous study [33]. 

Figure 2 represents the examples of the smoothed tri-axial acceleration data of unipedal balancing on a firm surface, a foam pad, and an MFT Challenge Disc plotted in a 3D coordinate space. For acceleration-based variable analysis, the middle 40 s of individual acceleration signals (anteroposterior (AP), mediolateral (ML), and vertical (VT) accelerations) were selected to discard movements associated with settling in or impatience at the end of the task [12]. 

### 2.5. Acceleration-Based Variable Computation

Sample entropy (SampEn) is a technique used to quantify the regularity of the signal [34] appropriately for short-length and noisy physiological data [35]. In order to formulate SampEn, the following main parameters are necessitated: the embedding dimension m (i.e., length of compared runs); the tolerance value r (i.e., similarity criterion); and the total length, N, of the analyzed time-series [34]. SampEn computes the probability that a sequence of data points, which has repeated itself within a tolerance r for a window length m, will also repeat itself for m + 1 points without allowing self-matches [35]. The SampEn algorithm was fully described by Estrada et al. [34].

The current study applied the SampEn to individual acceleration signals to measure the regularity of acceleration displacements, reflecting postural sway regularity, i.e., postural control complexity [35,36,37]. SampEn was calculated with three specific parameters in accordance with the previous studies [21,34]: embedding dimension m = 2, tolerance r = 0.2·STD, where STD is the standard deviation of the time series, and a time delay (τ) = 100 ms, which corresponds to physiological timescales. 

Moreover, two more acceleration-based variables were computed from individual acceleration signals: the root mean square (RMS) as a measure of the magnitude of an acceleration signal, i.e., the variability index of postural movements [38], and the 95% confidence ellipse area covered in the ML and AP directions as a measure of postural sway area, i.e., an index of the overall performance of postural control—the smaller the area, the better the performance [38]. 

### 2.6. Statistical Analysis

The SPSS software version 26.0 (IBM SPSS Statistics, SPSS Inc., Chicago, IL, USA) was used for all statistical analyses, with the alpha level set at a = 0.05. A Shapiro–Wilk test was used to test the normal distribution of the considered variables. A repeated-measures ANOVA was applied to acceleration variables to test the effects of two factors (leg dominance and surface stability). The effect sizes (Partial Eta Square; η_p_^2^) and observed power (1 − β) were also reported. The Holm–Bonferroni correction [39] was applied to adjust the alpha level to control the family-wise error rate (7 comparisons for each factor). 

## 3. Results

### 3.1. Leg Dominance Effects

All participants completed all the tests without losing their balance. The main results show that leg dominance effects are observed only in the SampEn variables (Table 2). Specifically, unipedal balancing by the dominant leg (DO) shows greater SampEn_VT (F_(1.0,21.0)_ = 84.98, *p* < 0.001, η_p_^2^ = 0.802, 1 − β = 1), SampEn_ML (F_(1.0,21.0)_ = 63.23, *p* < 0.001, η_p_^2^ = 0.751, 1 − β = 1), and SampEn_AP (F_(1.0,21.0)_ = 181.81, *p* < 0.001, η_p_^2^ = 0.896, 1 − β = 1) values than unipedal balancing by the non-dominant leg (ND).

### 3.2. Surface Stability Effects

The main results show surface stability effects observed in all acceleration variables. For the SampEn variables, surface stability effects appear in SampEn_VT (F_(1.81,38.17)_ = 14.32, *p* < 0.001, η_p_^2^ = 0.406, 1 − β = 0.998), SampEn_ML (F_(1.54,32.33)_ = 25.01, *p* < 0.001, η_p_^2^ = 0.544, 1 − β = 1), and SampEn_AP (F_(1.96,41.19)_ = 107.11, *p* < 0.001, η_p_^2^ = 0.836, 1 − β = 1). For the RMS variables, surface stability effects are observed in RMS_VT (F_(1.55,32.65)_ = 36.25, *p* < 0.001, η_p_^2^ = 0.633, 1 − β = 1), RMS_ML (F_(1.37,28.70)_ = 22.48, *p* < 0.001, η_p_^2^ = 0.517, 1 − β = 0.999), and RMS_AP (F_(1.47,30.88)_ = 21.35, *p* < 0.001, η_p_^2^ = 0.504, 1 − β = 0.999). Surface stability also has an effect on the 95% ellipse area variable (F_(1.15,24.17)_ = 181.81, *p* = 0.004, η_p_^2^ = 0.303, 1 − β = 0.860). 

The post hoc tests represented in Table 3 reveal the significant differences between balancing on different surfaces seen in the specific pairs of surface conditions. 

As shown in the first pair (Firm vs. Foam), balancing on a firm surface has greater SampEn_AP (*p* ≤ 0.001), but smaller RMS_VT (*p* ≤ 0.001), RMS_ML (*p* ≤ 0.001), RMS_AP (*p* ≤ 0.001), and 95% ellipse area (*p* = 0.001) values than balancing on a foam surface. 

Regarding the second pair (Firm vs. MFT), the significant differences between these two surface conditions are observed only in the SampEn, of which balancing on a firm surface has greater SampEn_VT (*p* ≤ 0.001), SampEn_ML (*p* ≤ 0.001), and SampEn_AT (*p* ≤ 0.001) values than balancing on an MFT board. 

In addition, the third pair (Foam vs. MFT) shows that the significant differences between these two surface conditions are observed only in the SampEn and RMS, a phenomenon which is seen in all acceleration directions. Specifically, balancing on a foam surface has greater SampEn_VT (*p* = 0.001), SampEn_ML (*p* = 0.001), SampEn_AT (*p* = 0.001), RMS_VT (*p* = 0.001), RMS_ML (*p* = 0.002), and RMS_AP (*p* ≤ 0.001) values than balancing on an MFT board.

### 3.3. Leg Dominance and Surface Stability Interaction Effects

The main results show that the interaction effects between leg dominance and surface stability are only observed in the SampEn variables: SampEn_VT (F_(1.88,39.53)_ = 145.82, *p* < 0.001, η_p_^2^ = 0.874, 1 − β = 1), SampEn_ML (F_(1.38,29.08)_ = 137.27, *p* < 0.001, η_p_^2^ = 0.867, 1 − β = 1), and SampEn_AP (F_(1.48,31.03)_ = 246.94, *p* < 0.001, η_p_^2^ = 0.922, 1 − β = 1). As shown in Figure 3, it is indicated that unipedal balancing on a firm surface influences the interaction effects. The high SampEn which is observed for the dominant leg is decreased when unipedal balancing is performed on both unstable surfaces (foam and MFT). 

## 4. Discussion

The current study investigated the effects of leg dominance, surface stability and their interactions on postural control during balancing on three different surfaces with increasing instability by assessing postural acceleration derived from a triaxial accelerometer-based smartphone sensor. Three acceleration-based variables—SampEn, RMS, and 95% ellipse area—were computed as measures of the regularity of postural sway, the magnitude of an acceleration signal, and the area of postural sway, respectively. The main results show that leg dominance effects are observed only in the SampEn, while surface stability effects appear in all acceleration-based variables. In addition, the interaction effects between leg dominance and surface stability emerge only in the SampEn data. Determined according to the current findings, the following three main points were discussed. 

Firstly, interlimb differences in postural control are shown as being more irregular postural sway fluctuations (high SampEn) which appear during balancing on the dominant (kicking) leg and which may reflect higher postural control efficiency or automaticity than balancing on the non-dominant leg. These findings support the hypothesis that the SampEn variable, a relevant index measuring a given acceleration time series’ variability (i.e., complexity), could reflect an inherent difference in neuromuscular control between legs. In addition, the current findings agree with the previous notion that measuring the postural accelerations could detect an inherent interlimb difference in postural control [8,9,10], reflecting the inherent bilateral asymmetry between the motor control circuitry of the two hemispheres [1]. However, as reported in a review study [7], the distinct mechanisms of the interlimb differences in postural control have not yet been explained. Further exploratory research will, therefore, be helpful to understanding the impact of leg dominance as one of the injury risk factors, especially for noncontact sports-related lower-limb injuries [3], for which asymmetry in the neuromuscular control has been suggested as one of the possible mechanisms [9]. 

Secondly, the surface stability effects observed in all acceleration-based variables indicate that increasing surface instability could increase the difficulty in controlling postural stability, which is typically seen in higher electromyographic (EMG) activities of the leg muscles when balancing on unstable surfaces than when balancing on a level, rigid surface [11,40,41]. As previously reported [42], balancing on unstable surfaces could modulate postural control by decreasing the ability to use ankle torque to maintain an upright posture while increasing the contributions of hip strategies and knee movements to doing so, which has alternatively been seen as having a greater sway than balancing on a stable surface. Moreover, it is suggested that increasing surface instability increases the requirement for neuromuscular control in order to achieve equilibrium [11]. When considering the sensory processes in balance control, the effective interaction among orientation inputs from somatosensory (proprioceptive, cutaneous, and joint), visual, and vestibular systems is required to achieve the given balance tasks [43]. It has been reported that balancing under threatening postural conditions could increase proprioceptive inputs, specifically from muscle spindle sensitivity [44,45], the muscle spindle being a major proprioceptive receptor sensitive to changes in muscle length during movements [46]. In other words, the greater the oscillations of the support surface, the higher the proprioceptive feedback [44]. In this sense, unipedal training on unstable surfaces has been suggested as an essential therapeutic exercise with proven effectiveness in facilitating and regaining neuromuscular control [11,40,47,48,49]. 

Thirdly, the interaction effects between leg dominance and surface stability in the SampEn reveal the more irregular postural sway fluctuations (higher SampEn) of the dominant (kicking) leg observed when unipedal balancing on the firm surface, which disappear when balancing on unstable surfaces. In this regard, the changes in SampEn that occurred when going from stable to unstable surfaces might reflect that infrequently balancing on unusual surfaces (foam pad and MFT Challenge Disc), faced daily, might alter the neuromuscular control needed to maintain balance. Interestingly, the current findings align with a previous study, which reported that interlimb differences in neuromuscular control disappeared when balancing on an unstable surface [9]. Based on the essential sensory processes involved in balance control, one possible reason might involve the upregulated somatosensory information, perceived by the central nervous system (CNS) and integrated into controlling movements and stability [46,50]. However, no report has shown how the neural systems control the two legs differently in processing sensory inputs and causing adaptative motor behavior. This point may be of interest to future studies.

In summary, the main empirical findings suggest that leg dominance effects could be effectively revealed by the nonlinear method (e.g., SampEn) when applied to postural acceleration and that increasing surface instability facilitates intended neuromuscular control. In addition, leg dominance and surface stability interactions, observed when measuring the regularity of postural acceleration signals, suggest that the exploitation of surface instability benefits should be considered to eliminate the effects of interlimb differences in postural control. Since leg dominance has been listed as one of the possible risk factors for sports-related lower-limb injuries [2,3,4], the current findings suggest that unipedal balancing training on unstable surfaces is recommended to reduce interlimb differences in neuromuscular control. Further examining postural acceleration in sports-related movements (e.g., jumping, turning, and landing) may provide more information for sports-related lower-limb injury prevention and rehabilitation.

## 5. Conclusions

The effects of leg dominance on postural control during unipedal balancing on three support surfaces with increasing instability were investigated using a nonlinear technique, namely sample entropy (SampEn), to measure the regularity of postural acceleration fluctuation. The effects of leg dominance and interactions between leg dominance and surface stability were observed in all acceleration directions. Balancing on the dominant leg showed more irregular postural acceleration variations (high SampEn), indicating greater postural control efficiency or automaticity than balancing on the non-dominant limb. In addition, the interaction effects supported the notion that unipedal balance on unstable surfaces may reduce interlimb differences in neuromuscular control.

## Figures and Tables

**Figure 1 sports-11-00075-f001:**
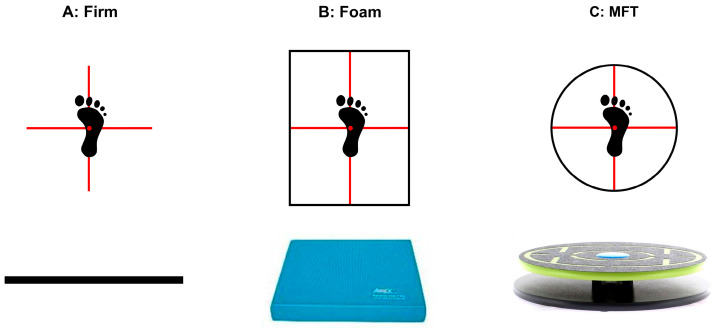
Illustrations of the standardized foot position on (**A**) a firm surface, (**B**) a foam pad, and (**C**) an MFT Challenge Disc.

**Figure 2 sports-11-00075-f002:**
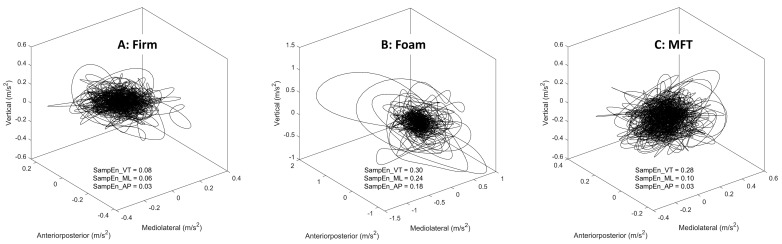
Example tri-axial acceleration data of unipedal balancing on (**A**) a firm surface, (**B**) a foam pad, and (**C**) an MFT Challenge Disc, with the corresponding SampEn calculated for each sway direction. Note: the presented data were retrieved from the first trial of one female participant during 40 s of unipedal balancing by the dominant leg on each support surface.

**Figure 3 sports-11-00075-f003:**
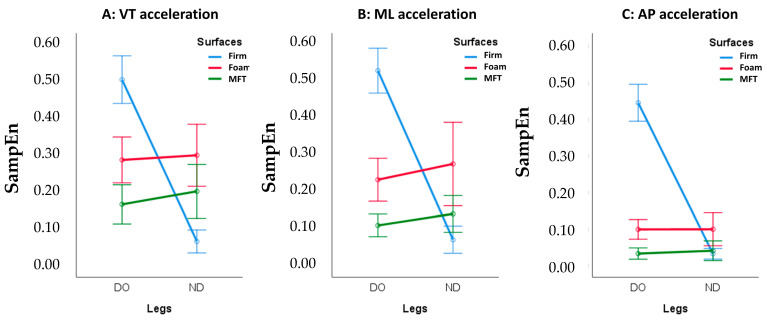
Interaction effects between leg dominance and surface stability on SampEn in (**A**) vertical (VT), (**B**) mediolateral (ML), and (**C**) anteroposterior (AP) accelerations.

**Table 1 sports-11-00075-t001:** Characteristics of participants (mean ± SD); * *p* < 0.001.

	Total (n = 22)	Male (n = 9)	Female (n = 13)	*p* Value
Age (years)	21.5 ± 0.6	21.3 ± 0.5	21.6 ± 0.6	0.282
Weight (kg)	58.2 ± 9.9	67.1 ± 5.8	52.0 ± 7.1	<0.001 *
Height (cm)	164.7 ± 9.6	174.6 ± 6.2	157.9 ± 3.7	<0.001 *
Body mass index (kg/m^2^)	21.2 ± 2.7	22.1 ± 1.6	20.6 ± 3.1	0.213

**Table 2 sports-11-00075-t002:** Comparing acceleration-based variables between the dominant (DO) and non-dominant (ND) legs (group mean ± Std. Error, * *p* < 0.001).

Variables	DO	ND	*p* Value	η_p_^2^	1 − β
SampEn_VT	0.32 ± 0.03	0.19 ± 0.03	<0.001 *	0.802	1
SampEn_ML	0.29 ± 0.02	0.16 ± 0.03	<0.001 *	0.751	1
SampEn_AP	0.19 ± 0.01	0.06 ± 0.01	<0.001 *	0.896	1
RMS_VT	0.12 ± 0.01	0.13 ± 0.02	0.290	0.053	0.179
RMS_ML	0.11 ± 0.01	0.12 ± 0.01	0.073	0.145	0.438
RMS_AP	0.07 ± 0.00	0.08 ± 0.01	0.497	0.022	0.101
95% ellipse area	0.17 ± 0.02	0.23 ± 0.07	0.225	0.069	0.223

**Table 3 sports-11-00075-t003:** Comparing acceleration-based variables between three surfaces: a firm surface, a foam pad, and an MFT Challenge Disc (group mean ± Std. Error). The symbol * denotes significance as determined by the satisfaction with the Holm–Bonferroni correction). Note: a = Firm vs. Foam, b = Firm vs. MFT, and c = Foam vs. MFT.

Variables	Firm	Foam	MFT	*p* Value ^a^	*p* Value ^b^	*p* Value ^c^	η_p_^2^	1 − β
SampEn_VT	0.28 ± 0.02	0.29 ± 0.03	0.18 ± 0.03	1.000	<0.001 *	0.001 *	0.534	0.983
SampEn_ML	0.30 ± 0.02	0.25 ± 0.04	0.12 ± 0.02	0.395	<0.001 *	0.001 *	0.826	1
SampEn_AP	0.24 ± 0.01	0.10 ± 0.02	0.03 ± 0.01	<0.001 *	<0.001 *	0.001 *	0.919	1
RMS_VT	0.09 ± 0.01	0.17 ± 0.01	0.12 ± 0.01	<0.001 *	0.012	0.001 *	0.881	1
RMS_ML	0.09 ± 0.01	0.16 ± 0.02	0.10 ± 0.01	<0.001 *	0.097	0.002 *	0.672	1
RMS_AP	0.07 ± 0.01	0.10 ± 0.01	0.06 ± 0.01	<0.001 *	1.000	<0.001 *	0.669	1
95% ellipse area	0.14 ± 0.04	0.34 ± 0.08	0.13 ± 0.02	0.001 *	1.000	0.033	0.588	0.996

## Data Availability

Not applicable.

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
