# Peer review of "Leg Dominance—Surface Stability Interaction: Effects on Postural Control Assessed by Smartphone-Based Accelerometry"

_sports, 2023, doi:10.3390/sports11040075_

Round 1

Reviewer 1 Report

The authors use an accelerometer based measurement to calculate the sample entropy of postural sway and investigate the effect of leg dominance and increasingly unstable surfaces. This is a very interesting and well-written study. My comments are only minor.

Introduction, line 31. Might be beneficial to specify particular injuries.

Introduction, lines 41-45. Is this a conclusion of the studies referenced or your own interpretation? If that latter, is it based on knowledge of your own results or an a priori assumption?

Introduction, lines 68-74. If SampEn is too high, the movements will become too irregular and random. Ideally, SampEn should be somewhere in the middle of its theoretical range, with deviations away from this in either direction reflective of poorer postural control.

Introduction, post line 74. It might be useful to elaborate on more previous research using SampEn to characterise postural control and to emphasise that the addition of SampEn to the measurements taken in this study is novel.

Lines 90 and 101. What does non-exercise physical activity mean?

Method, lines 131-136. Punctuation in this paragraph makes it seem like there are 4, rather than 3, conditions.

Method, line 141. Was the order of the firm, foam and MFT randomised?

Method, line 154. Were participants able to balance on each surface for the full 60 seconds? If they happened to lose balance and fall off during the trial, was it restarted?

Method, lines 180-191. Add here that SampEn was analysed for each direction of acceleration. Might be useful to add a little more background to how the SampEn calculation works as I suspect many people will be unfamiliar with it. It may even be useful to include the SampEn equation.

Method, lines 199-201. I am always wary of authors writing about statistical trends. If the values are not significant, they are not significant. I would suggest removing this sentence.

Results, lines 221-222. I would suggest writing about the results of the key post-hoc tests (i.e., the significant ones) here rather than simply refer readers to the table.

Results, lines 230-233. More description of these results is needed, rather than just referring to the figure.

Discussion, line 272. I thought the dominant leg had more irregular postural sway fluctuations?

Results and discussion. Whilst not one of the aims of your study, I think it might be interesting to mention/describe how SampEn actually changes when going from stable to unstable surfaces.

Author Response

Response to Reviewer 1 Comments

Point 1: The authors use an accelerometer-based measurement to calculate the sample entropy of postural sway and investigate the effect of leg dominance and increasingly unstable surfaces. This is a very interesting and well-written study. My comments are only minor.

Response 1: Thank you very much for your suggestions. I appreciated and recognized their values and applied them to improve the manuscript.

Point 2: Introduction, line 31. Might be beneficial to specify particular injuries.

Response 2: I added an example of the most common lower limb injuries (e.g., an anterior cruciate ligament (ACL) injury) influenced by leg dominance.

Point 3: Introduction, lines 41-45. Is this a conclusion of the studies referenced or your own interpretation? If that latter, is it based on knowledge of your own results or an a priori assumption?

Response 3: It is my own interpretation based on the findings of the previous studies. I rewrote those sentences a bit in order to make them clear.

Point 4: Introduction, lines 68-74. If SampEn is too high, the movements will become too irregular and random. Ideally, SampEn should be somewhere in the middle of its theoretical range, with deviations away from this in either direction reflective of poorer postural control.

Response 4: Thank you very much for your explanations. I added this part in the revised manuscript accordingly.

Point 5: Introduction, post line 74. It might be useful to elaborate on previous research using SampEn to characterize postural control and to emphasize that the addition of SampEn to the measurements taken in this study is novel.

Response 5: Thank you for the suggestions. In the revised manuscript, I added more previous studies using the SampEn to characterize postural control accordingly.

Point 6: Lines 90 and 101. What does non-exercise physical activity mean?

Response 6: The term “non-exercise physical activity” refers to non-sports or exercise participation. I also added more explanation about non-exercise physical activity in the revised manuscript.

Point 7: Method, lines 131-136. Punctuation in this paragraph makes it seem like there are 4, rather than 3, conditions.

Response 7: Thank you very much for pointing out the mistake. I rewrote those sentences to make them clearer in the revised manuscript.

Point 8: Method, line 141. Was the order of the firm, foam, and MFT randomized?

Response 8: Yes, the order of surface conditions is randomized. Thank you for pointing out the missing information. I added this information in the revised manuscript accordingly.

Point 9: Method, line 154. Were participants able to balance on each surface for the full 60 seconds? If they happened to lose balance and fall off during the trial, was it restarted?

Response 9: All participants could complete the tasks without losing their balance. However, this point was a concern in the study and had explained to each participant. In the revised manuscript, I added some explanations for the methods.

Point 10: Method, lines 180-191. Add here that SampEn was analyzed for each direction of acceleration. Might be useful to add a little more background to how the SampEn calculation works as I suspect many people will be unfamiliar with it. It may even be useful to include the SampEn equation.

Response 10: I added more explanations about the SampEn in the method. However, since there is an excellent presence of the SampEn equation in the previous study (Estrada et al. 2017) that the current study used as the reference, I only added the sentence to refer to that paper.

Point 11: Method, lines 199-201. I am always wary of authors writing about statistical trends. If the values are not significant, they are not significant. I would suggest removing this sentence.

Response 11: Thank you for your suggestions. I removed the sentences that you mentioned already.

Point 12: Results, lines 221-222. I would suggest writing about the results of the key post-hoc tests (i.e., the significant ones) here rather than simply refer readers to the table.

Response 12: Thank you for your suggestions. In the revised manuscript, I described the results of the post-hoc tests in the main text according to the suggestions.

Point 13: Results, lines 230-233. More description of these results is needed, rather than just referring to the figure.

Response 13: Thank you for your suggestions. In the revised manuscript, I described the results of the post-hoc tests in the main text according to the suggestions.

Point 14: Discussion, line 272. I thought the dominant leg had more irregular postural sway fluctuations?

Response 14: Thank you very much for pointing out my mistake. I checked and corrected that sentence already.

Point 15: Results and discussion. Whilst not one of the aims of your study, I think it might be interesting to mention/describe how SampEn actually changes when going from stable to unstable surfaces.

Response 15: Thank you very much for the suggestions. I added some possible explanations.

Reviewer 2 Report

Thank you very much for the opportunity to review this manuscript. The topic of the paper is interesting and fits the scope of the journal. The text is relatively well written and composed. The major limitation of this study is that the section of discussion is very poor. Please analyze more the discussion section. Also, the question that arises is why this study is important and what additional information offer? 

Author Response

Response to Reviewer 2 Comments

Point 1: Thank you very much for the opportunity to review this manuscript. The topic of the paper is interesting and fits the scope of the journal. The text is relatively well written and composed. The major limitation of this study is that the section of discussion is very poor. Please analyze more the discussion section. Also, the question that arises is why this study is important and what additional information offer? 

Response 1: Thank you very much for your comments and suggestions. I appreciate all the constructive suggestions and try my best to improve the manuscript based on your suggestions. Based on the suggestions, I rewrote the discussion section in the revised manuscript and hoped that this part would be clear.

Reviewer 3 Report

First, I would like to recognize the authors for the data they collected to investigate the effects of leg dominance on postural control during unipedal balancing on three different support surfaces with increasing instability.

The title is precise and accurate.

The abstract presents the rationale of the study and the general results are presented.

The introduction is clear and follows a logical sequence while all the relevant scientific support is provided. The presentation of leg dominance, postural control, postural acceleration, and differences between dominant and non-dominant legs when balancing on stable or unstable surfaces are accurately presented. The purpose and hypothesis are clearly presented.

The materials and methods section is presented with sufficient detail so that someone can replicate and build on the published results. The sample size is justified based on G*Power analysis. The use of an MFT disk makes this study unique for sports scientists who also use this disc.

Line 182-184: you can include your algorithm as well if you want. Did you create a formula? Not a must as you presented m, r, τ.

The results are accurately presented. Figure 3 is nicely presented.

The results were discussed, interpreted, and compared to previous studies in the discussion section.

Author Response

Response to Reviewer 3 Comments

Point 1: Line 182-184: you can include your algorithm as well if you want. Did you create a formula? Not a must as you presented m, r, τ.

Response 1: Thank you very much for your review. I appreciate all the constructive suggestions and try my best to improve the manuscript based on your suggestions. In the revised manuscript, I added more explanations about the SampEn in the method. However, since there is a nice presence of the SampEn equation in the previous study (Estrada et al. 2017) that the current study used as the reference, I only added the sentence to refer to that paper.

Round 2

Reviewer 2 Report

Paper accepted. Thank you for the good job.